# MInet: A Novel Network Model for Point Cloud Processing by Integrating Multi-Modal Information

**DOI:** 10.3390/s23146327

**Published:** 2023-07-12

**Authors:** Yuhao Wang, Yong Zuo, Zhihua Du, Xiaohan Song, Tian Luo, Xiaobin Hong, Jian Wu

**Affiliations:** School of Electronic Engineering, Beijing University of Post and Telecommunications, Beijing 100876, China; 1648541693@bupt.edu.cn (Y.W.); dzh2018010256@bupt.edu.cn (Z.D.); xhs@bupt.edu.cn (X.S.); luotian@bupt.edu.cn (T.L.); xbhong@bupt.edu.cn (X.H.); jianwu@bupt.edu.cn (J.W.)

**Keywords:** point cloud, LiDAR, multi-modal information, segmentation, object recognition

## Abstract

Three-dimensional LiDAR systems that capture point cloud data enable the simultaneous acquisition of spatial geometry and multi-wavelength intensity information, thereby paving the way for three-dimensional point cloud recognition and processing. However, due to the irregular distribution, low resolution of point clouds, and limited spatial recognition accuracy in complex environments, inherent errors occur in classifying and segmenting the acquired target information. Conversely, two-dimensional visible light images provide real-color information, enabling the distinction of object contours and fine details, thus yielding clear, high-resolution images when desired. The integration of two-dimensional information with point clouds offers complementary advantages. In this paper, we present the incorporation of two-dimensional information to form a multi-modal representation. From this, we extract local features to establish three-dimensional geometric relationships and two-dimensional color relationships. We introduce a novel network model, termed MInet (Multi-Information net), which effectively captures features relating to both two-dimensional color and three-dimensional pose information. This enhanced network model improves feature saliency, thereby facilitating superior segmentation and recognition tasks. We evaluate our MInet architecture using the ShapeNet and ThreeDMatch datasets for point cloud segmentation, and the Stanford dataset for object recognition. The robust results, coupled with quantitative and qualitative experiments, demonstrate the superior performance of our proposed method in point cloud segmentation and object recognition tasks.

## 1. Introduction

As autonomous driving and unmanned systems increase in application, the importance of environmental modeling, recognition, segmentation, and other processing tasks has escalated. Additionally, high spatial precision in complex environments has become vital, particularly for autonomous driving [1]. Consequently, using different sensors to acquire multi-modal information has emerged as a prevalent trend, with devices such as RGB (Two-dimensional color space information of the target) cameras [2], light detection and ranging (LiDAR) sensors for distance measurement [3], depth cameras, and radar sensors becoming increasingly common. LiDAR has established itself as a key element of 3D perception systems, enabling direct spatial dimension measurements [4] and yielding accurate 3D representations. Mobile point clouds derived from LiDAR are applicable to various tasks, such as object detection, tracking, semantic segmentation, and point cloud recognition [5]. Moreover, the geometric information encapsulated in point clouds is fundamental to numerous applications, including road network management, building and urban planning, and 3D high-definition mapping for autonomous vehicles [6].

At present, established processing algorithms for LiDAR point clouds encompass PointNet [7], PointNet++ [8], GACNet [9], DGCNN [10], among others. However, feature extraction in these algorithms only applies to the three-dimensional geometric features of the target object, which is not enough to fully characterize the object, which makes further point cloud processing still a considerable challenge. Thus, the application of suitable characteristics proved to be a research challenge. The primary obstacle in point cloud segmentation and recognition is the selection of more suitable features. To address this challenge, we propose MInet (Multi-Information net model), a network architecture for multi-modal information recognition. Our approach incorporates the 2D color information features of objects and their 2D color normals to enrich the interpretation of objects. Acknowledging the significance of point cloud normals [11]—the feedback of neighborhood support, or k-neighbors—which provide point positional information, we innovatively suggest the “normals” of point color information in 2D features. Our method involves using point cloud data that combines 3D positional information (XYZ) and 2D color space information (RGB) [12] as input, thereby integrating a multi-modal information point cloud structure. We apply our proposed MInet to the standard ShapeNet dataset [13] and the complex environment dataset, ThreeDMatch [14], for segmentation tasks, and the Stanford3D [15] dataset for recognition tasks.

The principal contributions of this research include:

(1) The introduction of a novel 2D image feature, RGB normals, which estimates the RGB normals at a point based on the RGB values from neighboring points. This feature functions as a rendering effect in the surrounding environment, thereby further enhancing the collection of color information from neighboring points.

(2) A novel end-to-end MInet for LiDAR point cloud recognition tasks is proposed in this study. The innovative aspect of this network is that it enhances the feature extraction of 2D data on the basis of 3D data, and integrates the features of both as multi-element information features for subsequent point cloud processing.

(3) Comprehensive comparisons investigating the feasibility of multi-modal information in point cloud processing technology, thereby demonstrating the superior performance of the proposed architecture.

In Section 2, we outline the latest domestic and international research developments, as well as the problems that this experiment aims to address. In Section 3, we primarily elaborate on the network architecture of the proposed Mlnet and its method of implementation, inclusive of a detailed explanation of the loss function utilized in this experiment. In Section 4, we undertake a comparative analysis of the proposed Mlnet and other models, scrutinizing their robustness in segmentation and recognition across various datasets. Section 5 provides a summative conclusion and future prospects. Here, we reflect on the work completed in this paper, articulate the innovative aspects of this study, evaluate the advantages and shortcomings of the proposed scheme, and indicate directions for future research endeavors.

## 2. Related Works

The pioneering PointNet model, brought forward by Charles R. Qi et al. at Stanford University [7], revolutionized the handling of point clouds by processing them directly. Each point in the input point cloud is assigned a corresponding spatial encoding, which are subsequently combined to create a global point cloud feature. This novel approach has not only achieved commendable results in point cloud processing, but also has made profound contributions to the broader field. Building on this, Qi went on to develop an enhanced version, PointNet++ [8], which innovatively introduces a multi-level feature extraction structure, facilitating the efficient extraction and comparison of local and global features.

Furthering the discourse in this field, Klokov and Lempitsky [16] presented a novel deep learning architecture, Kd-network, specifically tailored for 3D model recognition tasks involving unstructured point cloud data. By employing multiplicative transformations and kd-tree partitions with shared parameters, their approach deftly sidesteps the scalability concerns that plague traditional grid-based convolution architectures. In various tasks, including shape classification, retrieval, and part segmentation, the Kd-network has shown a performance competitive with established methods.

In a similar vein, Li et al. proposed PointCNN [17], an elegantly simple yet universally applicable framework for point cloud feature learning. By applying X-transformations for point reordering and weighting, followed by operations akin to convolution, PointCNN is capable of effectively managing unordered and irregular point cloud data, achieving performance that either matches or surpasses the most advanced methods across a multitude of tasks and datasets.

Addressing the potential loss of graph structural information during transitions and the amalgamation of redundant information, Bo Wu offered a solution with his Dynamic Graph CNN (DGCNN) [10]. This network expands upon the conventional CNN by integrating a preprocessing layer known as EdgeConv (DGCL). It further enhances convolution layers by implementing a backward propagation optimization process, thereby mitigating information loss in graph transformations. Similarly, to navigate the inherent constraints in point cloud semantic segmentation due to isotropy of its features, Lei Wang put forth a novel Graph Attention Convolution (GAC) [9]. This approach allows the kernel to be dynamically tailored to fit the structure of objects, enabling GACNet to extract structured features of point clouds for fine-grained segmentation while preventing inter-object feature contamination. Zhang proposed an SVM classifier based on color and shape features for target classification and recognition [18].

Despite these strides in point cloud processing, challenges persist when dealing with diverse complex environments or situations demanding higher accuracy, often requiring complex parameter tuning. This process is not only time-intensive but also involves considerable technical effort to generate desired processing results from filtered point cloud data. To address this, our study proposes a multi-information-based deep point cloud processing algorithm. The enhanced neural network model uniquely separates the 3D geometric and 2D color information of the target object from the input data, forming distinct point cloud features after processing. Empirical results indicate that the proposed multi-information processing approach demonstrates robust performance across various environmental conditions and requires minimal manual intervention. Consequently, our user-friendly and efficient method stands as a promising algorithmic model in this field.

## 3. Methods

This experiment capitalizes on multi-modal information for object feature extraction, and implements a novel local feature descriptor to enhance the accuracy and robustness of this extraction process. In this context, we propose a network model dubbed MInet, which accepts parallel inputs of multi-modal information. We utilize two network architectures to separately extract information from the three-dimensional and two-dimensional features of the target point cloud. In the following section, we will detail the feature extraction process for each dimension separately.

### 3.1. 3D Information Feature Extraction

Initially, the model ingests a dataset formatted as (N∗3), where the three-dimensional coordinates (x,y,z) of the target point cloud are fed into the model. Once the dataset is prepared, a sampling process is conducted. For this process, we use the SG module (sample and grouping module) to sample and group data sets. We employ the Farthest Point Sampling (FPS) [19] algorithm (i.e., the sampling process in the SG module of the flowchart in Figure 1), which affords superior coverage of the entire point set compared to other sampling algorithms. A total of $N^{\prime }$ points are selected as samples. To ensure the assembled sample points meet the criteria of compactness and rationality, we set $N^{\prime }=2048$ for segmentation tasks and $N^{\prime }=4096$ for recognition tasks. Subsequently, we group the sampled points. During this grouping process, we utilize the Ball Query method to generate $N^{\prime }$ local regions (i.e., the grouping process in the SG module of the flowchart in Figure 1), from which local features are extracted. Given the possibility of encountering non-uniform point clouds during point cloud processing, we employ a multi-scale grouping strategy for local feature extraction. For each central point, we operate at three different scales by constructing three regions around each central point. Each region boasts a unique radius and a different number of points. For any given central point, three disparate scale regions are processed with three sets of MLPs and convolution kernels, specifically (32, 64, 128), (64, 128, 256), and (64, 128, 256). Following this, all the local features are amalgamated into global features, as depicted in Figure 2. The global feature extraction process and the overall three-dimensional information feature extraction process are shown in Figure 1.

### 3.2. Two-dimensional Information Feature Extraction

Building on the foundation of two-dimensional information, we propose the utilization of color normal vectors to encapsulate the two-dimensional color information. Color normals, serving as a specific type of normal vector, embody the influence of the surrounding colors on a point’s color information. This study also carried out the process of obtaining color normal vector for two-dimensional color features. We interpret the task of normal vector estimation as the projection of a normal vector tangent to the surface. Consequently, the problem of normal vector estimation is reformulated as a least-squares plane fitting estimation problem. Through an examination of the eigenvectors and eigenvalues of a covariance matrix (also referred to as Principal Component Analysis–PCA [20]), we simplify this problem. The covariance matrix is generated by the nearest neighbors of the query point. For each point $p_{i}$, we define the covariance matrix *C* as follows:(1)$$C=\frac{1}{k}\sum_{i=1}^{k}\left(p_{i}-\bar{p}\right){\left(p_{i}-\bar{p}\right)}^{T},C\vec{v_{j}}=\lambda_{j}\vec{v_{j}},j\in \left\{0,1,2\right\}$$
where

*k* is the nearest k points to $p_{i}$ point;

$\bar{p}$ is the center of the nearest neighbor;

$\lambda_{j}$ is the *j*-th eigenvalue;

$v_{j}$ is the *j*-th eigenvalue vector.

Based on the neighborhood support estimation from the (r,g,b) values of each point, the color normal of the point, representing the normal vector of the two-dimensional color information, is determined. The designed structure for two-dimensional feature extraction is illustrated in Figure 3. It takes a set of points with $d_{\mathrm{in}}$ dimensions as input data. The input undergoes three different convolutional processes with kernel sizes of 16, 32, and 64, respectively. The results obtained from the three convolutions are then integrated, followed by the extraction of two-dimensional information features using MLP layers with three kernels of size 64, 32, and num_class. This process generates the final global feature representation. When only (r,g,b) information is available, $d_{\mathrm{in}}=3$. When both (r,g,b) and (r,g,b) color normals are considered, $d_{\mathrm{in}}=6$. In the flowchart shown in Figure 3, the entire process of two-dimensional information feature extraction for our experiment is presented.

### 3.3. Object Feature

In previous studies, two-dimensional features are often used to discriminate data processed with three-dimensional features. However, relying on a single-dimensional feature has proven to be inadequate in addressing all potential scenarios. Consequently, in this study, we propose a method that concurrently processes the target using both two-dimensional and three-dimensional features. This methodology bolsters the robustness of our experiment in terms of target segmentation and recognition. Based on this, our experiment proposes the process framework shown in Figure 4, combining two-dimensional and three-dimensional features to form unique point cloud characteristics. This lays the groundwork for subsequent point cloud processing tasks.

### 3.4. Loss Function and Evaluation

This model does not employ feature transformation matrices; hence, this study resorts to the cross-entropy loss function [21] for our experimental loss function. Cross-entropy serves as a measure to gauge the divergence between two distinct probability distributions over the same random variable. Within the context of machine learning, it quantifies the disparity between the true probability distribution and the predicted probability distribution. The expression for the loss function is provided as follows:(2)$$H\left(p,q\right)=-\sum_{i=1}^{n}p\left(x_{i}\right)\log\left(q\left(x_{i}\right)\right)$$

In order to provide a quantitative comparison and analysis of the classification results for LiDAR point clouds, we employ several evaluation metrics. These include overall accuracy (*OA*) [22], mean intersection over union (*mIoU*) [23], *recall* [24], *precision* [25], and the *f*_1_ score [26].

The metrics are defined as follows:(3)$$OA=\frac{tp+tn}{tp+tn+fp+fn}$$
(4)$$mIoU=\frac{tp}{tp+fp+fn}$$
(5)$$precision=\frac{tp}{tp+fp}$$
(6)$$recall=\frac{tp}{tp+fn}$$
(7)$$f_{1}\text{-}\mathrm{score}=2*\frac{precision*recall}{precision+recall}$$

All the experiments were implemented in the ubuntu18.04 environment using a single GPU. A total of 200 epochs were used with 0.1 momentum and 0.001 initial learning rate as training parameters. Set batch_size to 8 for hardware performance. i7-12700, 2.10 GHz processor and RTX 3080 Ti graphics card were used in the experiment. Results were evaluated by overall accuracy (*OA*), average crossover and association (*mIoU*), *recall*, *precision*, and *f*_1_ scores.

## 4. Result and Discussion

### 4.1. Data Introduction

To obtain the requisite multi-modal information, we blend two-dimensional images and three-dimensional point clouds. Rapid acquisition of three-dimensional point clouds is possible through LiDAR, while visible light cameras supply the two-dimensional images. Moreover, training datasets that sufficiently represent complex environments are necessary. Summarizing, we have chosen the ShapeNet [13] and 3DMatch [14] datasets as input data to appraise the segmentation performance of our model. ShapeNet, developed by Princeton University, is a large-scale 3D model dataset encompassing various model types, including furniture, animals, vehicles, and buildings. Each 3D model is supplemented with metadata such as classification labels, poses, and materials. Distinctly, ShapeNet segregates geometric and semantic information, permitting researchers to utilize these models conveniently for specific problems. 3DMatch is an extensive dataset collaboratively developed by researchers from ETH Zurich, Stanford, and Princeton University. Comprising pairs of 3D point clouds from three disparate sensors, it is employed to evaluate the effectiveness of various point cloud matching algorithms. This dataset finds wide applications in fields like 3D reconstruction, SLAM, map creation, and robot navigation. Moreover, we employ the Stanford [15] standard dataset to verify our model’s recognition performance. This dataset, commonly used for more demanding semantic scene segmentation tasks, consists of six indoor areas with 2.79 billion points scanned from 271 rooms across three different buildings. All points are labeled with 13 semantic categories, including board, chair, and other common objects. Rooms are subdivided into 1 m × 1 m blocks. We represent the point clouds using a nine-dimensional vector ((x,y,z), (r,g,b), and normalized coordinates).

### 4.2. Experiment Result

#### 4.2.1. Result of Shapenet Dataset Segmentation

In our study, we performed a rigorous evaluation of our network’s performance on the ShapeNet dataset using well-established metrics, namely, category-average Intersection over Union (*mIoU*) and instance *mIoU*, facilitating direct comparison with other state-of-the-art methods. We set the bar high, benchmarking our network against a wide range of robust models such as PointNet, PointNet++, DGCNN, RGCNN [27] and PCT [28]. The empirical results obtained by MInet and other competing models were graphically visualized to enable an intuitive and direct assessment of their performance. Table 1, provides a concise yet comprehensive snapshot of the part segmentation results produced by our network in comparison to other models. Notably, our results underscore the proficiency of our network in extracting and encapsulating local information, especially when handling multi-faceted data. This is in stark contrast to other methodologies that confine themselves to the acquisition of three-dimensional information.

To compare the various models, we evaluated the results yielded by other methodologies, and juxtaposed them against the results obtained from MInet.The visual representation of the trained data of MInet and pointnet++ is shown in Figure 5. A noteworthy observation is the visually apparent discrepancy in the airplane segmentation results in the last row. PointNet++ exhibited minor shortcomings in accurately segmenting the airplane wings, whereas our method demonstrated a distinct ability to segregate all the parts of the airplane accurately. Moreover, the inability of PointNet++ to incorporate a two-dimensional representation of the objects resulted in an indistinct separation of the backrest and legs of the chair in the chair segmentation scenario presented in the second row. In stark contrast, our method leverages a synergistic combination of two-dimensional and three-dimensional information to seize the most critical features of the chair, yielding accurate segmentation results. The fidelity of our results, the true value of the data entered, coupled with a thorough comparison of the visualization results, unequivocally substantiates the superior accuracy and robustness of our network. This claim is further reinforced visually in Table 1, which exhibits the high *mIoU* value of 89.3% achieved by our network model, thereby further showcasing its efficacy and precision.

#### 4.2.2. Result of 3DMatch Dataset Segmentation

For 3DMatch data in a complex environment, we added the normal vector of two-dimensional color information into the input data, and conducted training tests on MInet, pointnet++, pointnet, dgcnn and gacnet using the specified training parameters on the 3DMatch data set.

Table 2 presents the segmentation outcomes for different networks on the 3DMatch dataset. An analysis of the training and evaluation of several methods—MInet, PointNet, PointNet++, DGCNN, and GACNet—suggests that the MInet approach proposed in our study boasts the highest level of robustness. Notably, its accuracy demonstrates an enhancement of 18.8% and 17.9% in comparison with PointNet++ and DGCNN, respectively. Figure 6 visualizes the comparison between the training values generated by MInet and the ground truth values, revealing superior performance of our trained model across various complex environments, such as living rooms, staircases, and bedrooms. This superiority is predominantly attributed to MInet’s focus on the multi-faceted information of the target object. In complex settings, a singular focus on three-dimensional geometric data is insufficient to fully represent the target object. However, by incorporating multi-informational aspects, such as color data and color normals, we can extract more specific and detailed target features.

During the testing phase, we observed that our model performed best in environments with a wide range of colors, and it maintains high robustness even in environments with minimal color variations. Table 3 and Table 4, respectively, offer examples of four types of targets within environments featuring larger and smaller color spans, reinforcing the versatility and adaptability of our approach.

#### 4.2.3. Result of Stanford 3D Dataset Recognition

In our approach to model training, we strategically extracted 4096 points at random from each block. Notably, the test points were spared from any sampling process. We deployed a comprehensive six-fold cross-validation, gauging the efficacy of our model based on the average *mIoU* results and the Overall Accuracy (*OA*). Our methodology faithfully followed the same protocols of random scaling, rotation, and point expansion. The visualizations demonstrating the Stanford 3D test images can be viewed in Figure 7.

In Figure 7, we show a sharp contrast between MInet and PointNet++, which can highlight the advantages of the model proposed in this experiment in comparison to the actual situation. Careful examination of the selected examples—roof, podium and table—shows that our MInet is superior to existing models today. This advantage is clearly demonstrated in the roof recognition example depicted in the first row, where PointNet++ falls short in identifying the lights and four gray bricks on the roof. On the contrary, our model offers a distinct recognition result, primarily due to its enhanced sensitivity to variations in color regions. Moreover, in the bottom row that showcases table recognition, our model exhibits a sophisticated understanding of the varied colors of the tablecloth, while PointNet++ lags slightly behind in this regard. The superior performance of our proposed MInet is visually corroborated by the data presented in Table 5, where it outshines other models in terms of both *mIoU* and *OA*, registering scores of 57.2% and 87.1%, respectively. This underscores the effectiveness of our multi-information approach in handling complex point cloud recognition tasks.

## 5. Conclusions

In this research, we introduced a novel model named MInet, which assimilates multi-modal information extraction techniques based on a multi-spectral LiDAR system. This model is designed for the execution of point cloud segmentation and recognition tasks within diverse environments. MInet can effectively capture the local geometric relationship between the points obtained from the non-uniformly sampled data, and enhance the extracted features by loading multiple information and utilizing the two-dimensional and three-dimensional characteristics of the target object. The evaluation results elucidate that our MInet attains a *mIoU* of 0.893 on the ShapeNet dataset, while demonstrating recognition performance metrics of 0.855 *OA*, 0.922 *mIoU* on the 3DMatch dataset. Furthermore, the experimental outcomes showcase MInet’s superiority over other models in recognition tasks, in the semantic segmentation results of the Stanford 3D dataset, 0.572 of *mIoU* and 0.871 of *OA* were obtained.

MInet possesses the ability to extract abundant two-dimensional information from images while maintaining three-dimensional coordinate data, considerably facilitating semantic segmentation. This model introduces an innovative approach of utilizing multi-modal information to process targets, employing both two-dimensional and three-dimensional features. This breaks away from the prevailing practice of using two-dimensional features solely as a discriminator for three-dimensional ones [31]. In practical applications, most data collection and processing efforts employ multiple sensors. Our Mlnet, through the use of multi-modal information for data processing, aligns well with multi-sensor data collection and further enhances the accuracy of segmentation and recognition. Looking ahead, our objective is to extend our work to larger datasets and a wider variety of scenarios. This endeavor is aimed at ensuring that our methods are more comprehensive and efficient in real-world applications.

## Figures and Tables

**Figure 1 sensors-23-06327-f001:**
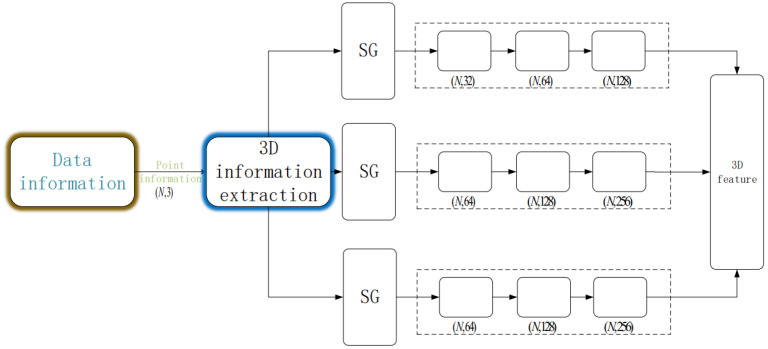
Three-dimensional feature extraction diagram. The SG module is FPS sample and grouping. (For three different scale regions of the same central point, each of the three sets of MLP, the convolution kernel is (32, 64, 128), (64, 128, 256), and (64, 128, 256), and then the three-dimensional features are obtained by concat operation).

**Figure 2 sensors-23-06327-f002:**
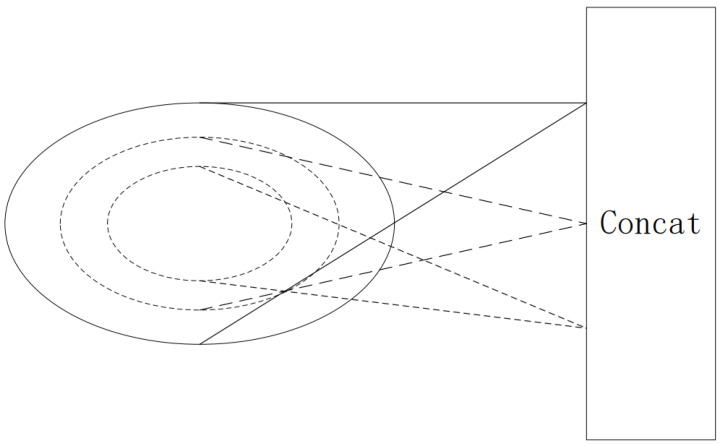
Local feature extraction (The local features are obtained by extracting features from three different scales).

**Figure 3 sensors-23-06327-f003:**
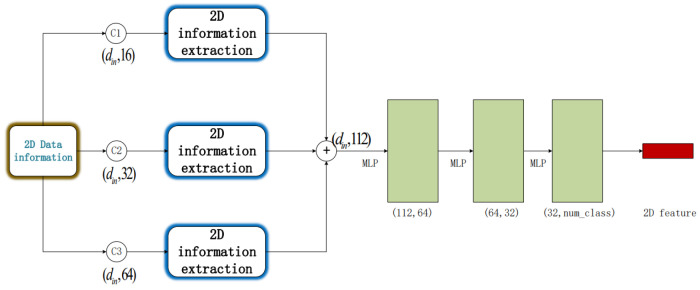
Two-dimensional feature extraction diagram (Three different convolution processes were carried out on the input, and the convolution kernels were 16, 32, and 64, respectively. After concat operation was carried out on the results obtained from the convolution processing, three MLP layers with convolution kernels 64, 32, num_class were used to extract two-dimensional information features and obtain the final two-dimensional features).

**Figure 4 sensors-23-06327-f004:**
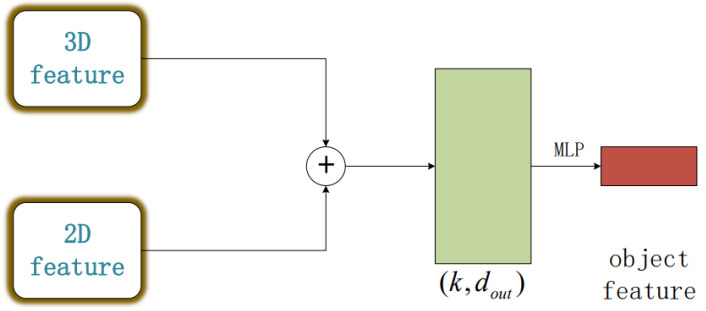
Object feature extraction (The multivariate features of the target are obtained through concat and MLP layers).

**Figure 5 sensors-23-06327-f005:**
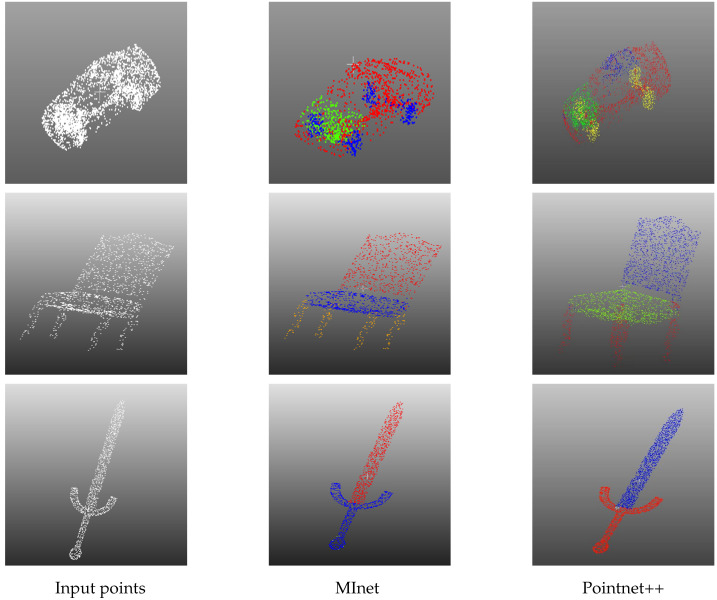
Visualization of segmentation results based on ShapeNet.

**Figure 6 sensors-23-06327-f006:**
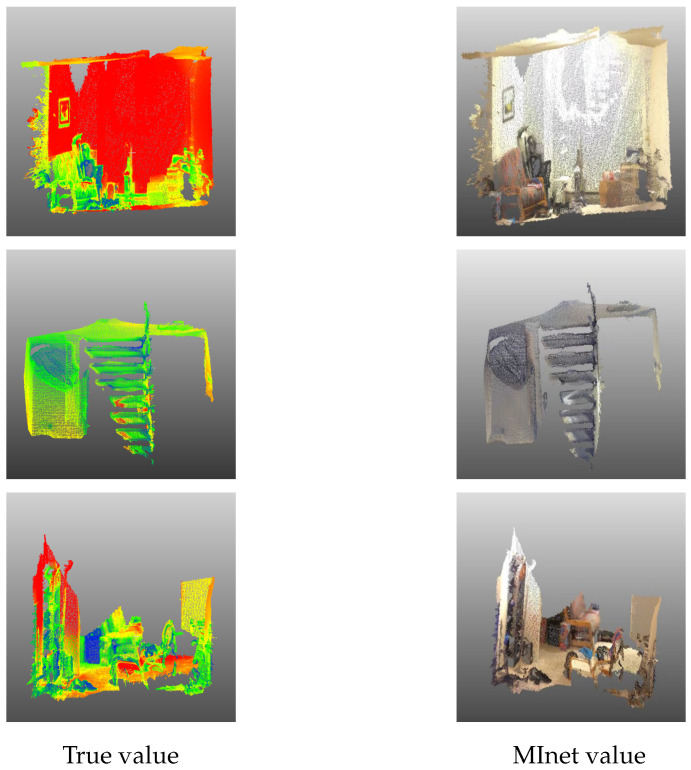
Visualization of segmentation results based on 3Dmatch.

**Figure 7 sensors-23-06327-f007:**
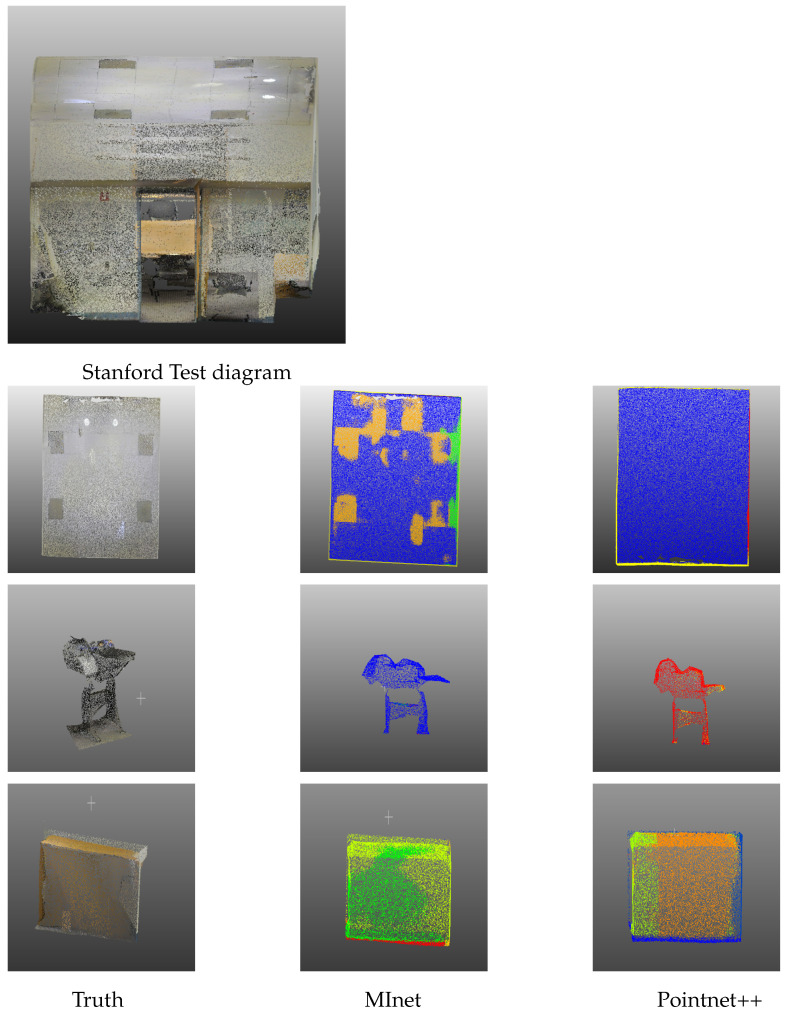
Visualization of different methods on S3DIS, from left to right: True value, MInet, Pointnet++.

**Table 1 sensors-23-06327-t001:** Segmentation results based on ShapeNet data (%).

Model	Miou	Plane	Bag	Cap	Car	Chair	Ear Phone	Guitar	Knife	Lamp	Laptop	Motor Bike	Mug	Pistol	Rocket	Skate Board	Table
shapes	—	2690	76	55	898	3758	69	787	392	1547	451	202	184	283	66	152	5271
Pointnet	82.5	82.2	77.6	81.3	73.9	88.3	72.0	90.2	84.7	79.7	94.0	64.3	91.7	80.1	57.1	71.8	79.5
Point net++	84.5	81.5	76.1	87.5	76.9	89.9	74.0	90.5	85.6	83.6	95.3	68.2	94.6	81.4	60.6	74.6	82.2
RGCNN	83.8	79.8	82.3	92.1	74.8	88.7	73.2	90.8	87.9	82.8	95.5	63.4	95.2	60.4	44.1	72.4	79.9
DGCNN	84.7	83.5	82.9	86.2	77.3	90.1	74.3	90.7	87.0	82.3	95.2	65.8	94.4	80.6	63.0	74.0	85.1
PCT	85.8	82.8	77.4	88.8	78.2	91.2	75.3	91.8	86.9	84.9	96.6	69.5	95.9	82.7	61.9	75.9	83.5
MInet	89.3	93.9	87.6	92.6	97.1	95.6	73.1	97.7	92.4	91.6	97.6	88.3	95.7	88.1	71.4	79.4	79.7

**Table 2 sensors-23-06327-t002:** Parameter comparison of training 3DMatch data of different models (%).

Model	Accuracy	*mIoU*	*Recall*	*Precision*	*f*_1_-Score
Pointnet	71.6	81.4	80.4	81.5	76.7
Pointnet++	72.0	80.5	76.7	78.5	79.2
Gacnet	72.5	85.7	77.9	82.6	78.7
Dgcnn	72.4	83.3	73.9	73.9	73.9
MInet	85.5	92.2	89.1	89.1	89.0

**Table 3 sensors-23-06327-t003:** The *mIoU* values of four types of objects in the environment with large color span (%).

	BDapt2	Browncs2	Chairs	BDcopyroom
Pointnet++	71.1	99.6	88.0	93.5
Gacnet	68.2	96.7	83.1	92.9
Dgcnn	75.0	97.5	80.0	92.9
MInet	99.8	99.6	97.8	97.9

**Table 4 sensors-23-06327-t004:** The *mIoU* values of four types of objects in the environment with small color span (%).

	Redikitchen	Pumpkin-Off	Office	BDoffice1
Pointnet++	71.6	73.9	79.5	83.4
Gacnet	67.7	65.4	68.3	70.4
Dgcnn	66.0	67.1	65.9	71.4
MInet	80.7	76.2	81.8	85.7

**Table 5 sensors-23-06327-t005:** Comparison results of model recognition.

Model	*mIoU* (%)	*OA* (%)
Pointnet	42.7	78.1
Pointnet++	52.3	82.2
Dgcnn	54.8	83.3
G+RCU [29]	44.6	80.7
AGnet [30]	58.2	85.1
MInet	57.2	87.1

## Data Availability

Not applicable.

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
