# Peer review of "MInet: A Novel Network Model for Point Cloud Processing by Integrating Multi-Modal Information"

_sensors, 2023, doi:10.3390/s23146327_

Round 1

Reviewer 1 Report

1. Please elaborate on the working principle or innovative aspects of MInet in lines 57-58.

2. In line 108, when referring to "a novel local feature descriptor," are you referring to RGB normals? To avoid confusion, please directly state the name of this descriptor.

3. In line 115, is the asterisk (*) indicating a multiplication symbol? If so, please replace it with the multiplication symbol. The definition of the dataset is unclear, please provide a clear definition.

4. Please provide a detailed discussion on related work regarding point cloud and image fusion in the related work section.

5. The text in the figures is too small, please enlarge it.

6. The figure captions in the text lack detailed descriptions of the content within the figures.

7. The full name of SG is not provided when it is first mentioned in the text.

8. In line 118, why is FPS mentioned before describing the SG module in the method flow? It would be more logical to first describe the SG module.

9. The article provides the value of N' in the method section, but what value is N set to?

10. What are the differences between the method described in section 3.1 and the SA method proposed in PointNet++?

11. The flowchart in Figure 2 provides a vague description of the algorithm. What does the white empty rectangle represent? What are the dimensions of the 3D feature?

12. What is the kernel of the MLP mentioned in line 158?

13. What do C1, C2, and C3 represent in Figure 3? Is 2D information extraction referring to CNN? What operation does the "+" symbol represent in the figure?

14. Please provide the definition of 2D information.

15. Please expand section 3.3 to explain how the two-dimensional and three-dimensional information is combined.

16. The definition of the loss function should be included in the method section.

17. Concepts such as OA and mIoU are widely known and do not need to be redefined.

18. It seems that the Shapenet dataset does not provide image information. Please provide a detailed explanation of how this work obtains 2D information.

19. What is RGCNN mentioned in Table 1?

20. In the experimental section, this work seems to only compare with methods proposed before 2020. Please add comparisons with more advanced models, especially those from recent years.

21. This work does not compare with any fusion methods. It is recommended to add comparisons with methods that combine images and point clouds to demonstrate the superiority of MInet.

Moderate editing of English language required

Reviewer 2 Report

The presented paper is in good shape in terms of scientific soundness and quality of presentation, 

I have very minimal remarks:

At the end of the introduction, a separate paragraph is required to describe the layout of the paper.

Line 26: Define RGB  

Please make Figs 2, 3 and 4 clearer 

Numbered every equation used included in the manuscript. 

Reviewer 3 Report

My comments:
1.
This paper is performed very well and has a certain contribution to related research fields.

2. I suggest to separate “Result” and “Discussion” into two sections. The authors have to strengthen them even more, because they are the core of a paper.

3. The section of Conclusion” must be reinforced more. For example, the contributions to academic research as well as practical application.

Minor editing of English language required

Reviewer 4 Report

Dear Author

The manuscript presents an up-to-date theme with relevance to the scientific community.

1- The abstract is well written and clear.

2 - The research methodology is credible and allows others to

3 - The conclusions are direct and communicate the findings found in the investigation.

4 - The bibliography must be revised so that master's dissertations are avoided in the context of a research article.

5 - The bibliography is very limited, leaving no mention of other non-Chinese publications of greater relevance, a situation that would be recommended to be reviewed.

Round 2
